# Micro-Elimination of Hepatitis C among Patients with Kidney Disease by Using Electronic Reminder System—A Hospital-Based Experience

**DOI:** 10.3390/jcm11020423

**Published:** 2022-01-14

**Authors:** Pei-Yuan Su, Wei-Wen Su, Yu-Chun Hsu, Shu-Yi Wang, Ping-Fang Chiu, Hsu-Heng Yen

**Affiliations:** 1Division of Gastroenterology, Department of Internal Medicine, Changhua Christian Hospital, Changhua 500, Taiwan; 35301@cch.org.tw (W.-W.S.); 77149@cch.org.tw (Y.-C.H.); 91646@cch.org.tw (H.-H.Y.); 2Division of Endocrinology and Metabolism, Department of Internal Medicine, Changhua Christian Hospital, Changhua 500, Taiwan; 86761@cch.org.tw; 3Division of Nephrology, Department of Internal Medicine, Changhua Christian Hospital, Changhua 500, Taiwan; 68505@cch.org.tw; 4Artificial Intelligence Development Center, Changhua Christian Hospital, Changhua 500, Taiwan; 5College of Medicine, National Chung Hsing University, Taichung 400, Taiwan

**Keywords:** electronic reminder system, HCV, screening, kidney disease

## Abstract

Background: Little is known about the use of an electronic reminder system for HCV screening among patients with kidney disease. In this study, we tried to determine whether reminder systems could improve the HCV screening rate in patients with kidney disease. Methods: Patients with kidney disease were enrolled from August 2019 to December 2020 to automatically screen and order HCV antibody and RNA testing in outpatient departments. Results: A total of 19,316 outpatients with kidney disease were included, and the mean age was 66.5 years. The assessment rate of HCV antibody increased from 53.1% prior to the reminder system to 79.8% after the reminder system (*p* < 0.001), and the assessment rate of HCV RNA increased from 71% to 82.9%. The anti-HCV seropositivity rate decreased from 7.3% at baseline to 2.5% after the implementation of the reminder system (*p* < 0.001), and the percentage of patients with detectable HCV RNA among those with anti-HCV seropositivity decreased from 69.1% at baseline to 46.8% (*p* < 0.001). Conclusions: The feasibility of an electronic reminder system for HCV screening among patients with kidney disease in a hospital-based setting was demonstrated.

## 1. Introduction

Hepatitis C is one of the major causes of cirrhosis and hepatoma. According to a WHO report, approximately 71.1 million persons experienced hepatitis C virus (HCV) viremia worldwide in 2017 [1]. Hepatitis C can induce liver-associated disease and cause many extrahepatic diseases, including insulin resistance, type 2 diabetes, cardiovascular disease, renal insufficiency, cryoglobulinemia, lymphoma, and neurological disorders [2]. The prevalence of HCV infection is higher in patients with chronic kidney disease and hemodialysis [3]. HCV can also accelerate the progression of chronic kidney disease (CKD) to end-stage renal disease (ESRD) and is associated with increased cardiovascular comorbidities in such patients [4,5,6]. The Kidney Disease: Improving Global Outcomes (KDIGO) guidelines published in 2018 suggest that all patients with CKD should be screened for HCV [7]. Little is known about the HCV prevalence rate in patients with kidney disease, including acute, chronic, or structural kidney diseases, and we designed the study to extend the screening population to all patients with kidney disease at outpatient departments.

There are many impediments to the HCV screening/treatment cascade, and the first barrier is the diagnosis of HCV [8]. An efficient method of improving HCV screening is essential to eliminate HCV. Automatic computerized alerts are the optimal method in clinical practice [9]. Electronic reminder systems have shown promising outcomes in Japan and the United States [10,11]. Alert systems have been used for populations with a high prevalence (i.e., baby boomers or those with HIV/HCV coinfection) in the United States, and the screening rates have increased. Limited data are available in relation to the use of an electronic reminder system for HCV screening in patients with kidney disease. We designed and assessed an electronic reminder system for HCV screening in patients with kidney disease in the outpatient departments of a medical center.

## 2. Materials and Methods

### 2.1. Materials

Outpatients with kidney disease, except those with end-stage renal disease (ESRD), were initially identified at baseline according to diagnoses by ICD-10-CM codes from previous medical records. Initially, we retrospectively identified patients with kidney disease, excluding end-stage renal disease (ESRD), according to diagnoses by ICD-10-CM codes from previous medical records, in the period of 1 January 2018 to 31 December 2018, at Changhua Christian Hospital before screening. These clinical data were recorded from the same population throughout the screening period. The study participants included patients regardless of whether they had anti-HCV antibody and HCV RNA data before screening. During the screening period, the participants received HCV antibody and RNA screening by computerized automatic medical record alerts during the same healthcare visit at an outpatient department. The inclusion criteria for kidney disease were defined by the following ICD-10-CM codes: A18.11, A52.75, C64, C7A.093, E11.2, E13.2, E10.2, E11.65, E10.65, E74.8, M10.3, N20.0, D59.3, I70.1, I75.81, I72.2, I77.73, I77.3, K76.7, N00–N08, N14–N15, N17, N18.1–4, N25, N26.9, O10.2–4, Q61, Q62, and R94.4. Patients with stage 5 chronic kidney disease or end-stage renal disease (ESRD) (ICD-10-CM codes: N18.5 and N18.6) were excluded because these patients at Changhua Christian Hospital had completed HCV screening and treatment before this study [12].

### 2.2. Alert System Implementation

Once a patient with kidney disease visited a physician in a primary care clinic, the electronic reminder system automatically searched the laboratory database for serum anti-HCV antibody test and HCV RNA test results from the past 10 years. Then, the reminder system was divided into two screening steps. In the first step, if the serum anti-HCV antibody test results were unavailable in the database, the computerized reminder system sent an alert message to the physicians and automatically ordered the anti-HCV test. The physicians could agree to the testing order or refuse the order if the patient refused or had received testing at another hospital. In the second step, for anti-HCV seropositive patients with unavailable serum HCV RNA test results, the reminder system automatically sent a pop-up message to the physician and ordered the serum HCV RNA quantification test during the same visit. A flowchart of the reminder system is shown in Figure 1.

The data were collected from the screening period (August 2019 to December 2020). The assessment rate of anti-HCV antibody is defined as the number of patients with available anti-HCV antibody data at prescreening (baseline) and post-screening divided by the number of eligible patients (the two denominators are the same as we defined at baseline). The assessment rate of HCV RNA was defined as the number of patients with available HCV RNA data at prescreening and post-screening divided by the number of patients who were anti-HCV seropositive at prescreening and post-screening (the two denominators were different because new anti-HCV seropositive patients were added after screening). The treatment rate was defined as the number of patients that completed antiviral therapy divided by the number of patients who had HCV RNA viremia at prescreening and post-screening (the two denominators were also different before and after screening). The Ethics Committee of Changhua Christian Hospital approved the study protocol used in this study (CCH IRB No: 200403), and informed consent was waived because the data were all anonymized.

### 2.3. HCV Testing

Anti-HCV antibodies were tested using an ARCHITECT anti-HCV assay (Abbott Laboratories, Abbott Park, IL, USA). HCV RNA was quantitatively measured by ART HCV assays (RealTime HCV test, Abbott Molecular, Abbott Park, IL, USA). The multistep testing procedure was adopted for the first 10 months of the study period, and then a combined approach with HCV reflex testing was used [13].

### 2.4. Statistical Analysis

We used Cochran’s Q test to compare the screening rate among the baseline and different screening phases and a post hoc test, Dunn’s test, which applies the Bonferroni correction, to compare each phase. A t test was utilized to compare age, and chi-square tests were used to compare sex, anti-HCV antibody levels and detectable HCV RNA at baseline and during the screening period. Statistical analyses were performed using PASW Statistics version 18 (formerly SPSS; IBM, Hong Kong). Statistical significance was defined as *p* < 0.05 (two-tailed).

## 3. Results

The total number of patients with kidney disease who were identified at baseline according to medical record diagnoses was 19,316 in August 2019, the mean age was 66.5 years old, and the population was predominantly male (55.6%).

### 3.1. HCV Screening Rate

The assessment rate of anti-HCV antibody increased from 53.1% prior to implementing the reminder system to 79.8% after the implementation of the reminder system. The screening rate of anti-HCV antibody at prescreening and post-screening was significantly different (*p* < 0.001) (Figure 2). The assessment rate of HCV RNA was 71% before implementing the reminder system and increased to 82.9% after the reminder system was implemented. The treatment rates were 85.8% and 84.7% before and after the implementation of the reminder system, respectively.

### 3.2. HCV Cascade of Care

The results of the HCV cascade of care of screening patients are shown in Figure 3. The number of patients with kidney disease who had automatically received orders for anti-HCV testing at outpatient clinics was 5538 during the screening period. In this group, the number and percentage of completed anti-HCV assessments, patients with anti-HCV seropositivity, patients who underwent HCV RNA testing, and patients with HCV viremia were 5154 (93.1%), 127 (2.5%), 111 (87.4%) and 52 (46.8%), respectively. A total of 41 (78.8%) viremia patients accepted direct-acting antiviral agents (DAAs), the intention to treat sustained virologic response rate (ITT-SVR) was 92.7%, and the per-protocol SVR rate (PP-SVR) was 100%.

### 3.3. HCV Testing Results

Table 1 shows the age, sex, HCV seropositivity rate and HCV RNA viremia rate among the patients who already had laboratory tests at baseline and among the patients who completed tests during the screening period. After HCV screening, the anti-HCV seropositivity rate was 5.7%, and the HCV RNA viremia rate was 65.7% in all patients, including the baseline and screening groups. The mean age was not significantly different between the two groups, but the baseline group was predominantly male compared with the screening group (56.7% vs. 51.4%, *p* < 0.001). The anti-HCV seropositivity rate decreased from 7.3% in the baseline group to 2.5% in the screening group (*p* < 0.001). The percentage of patients with detectable HCV RNA among anti-HCV seropositive patients decreased from 69.1% in the baseline group to 46.8% in the screening group (*p* < 0.001).

### 3.4. HCV Prevalence in Different Age Groups

The anti-HCV seropositivity rates and HCV RNA viremia rates in each age group are shown in Figure 4. Both rates were higher in older patients. The highest anti-HCV seropositivity rate was 7.0% in patients older than 90 years, and no patients younger than 30 years old were seropositive.

## 4. Discussion

This study is the first regarding the implementation of an electronic reminder system for HCV screening among outpatients with kidney disease. The overall screening rate for anti-HCV antibody increased from 53.1% to 79.8% in the 16-month screening period, and the assessment rate of HCV RNA increased from 71% to 82.9%. These results demonstrate the feasibility and usefulness of an electronic reminder system for HCV screening in a hospital-based setting.

Electronic reminder systems have been used for hepatitis screening for many years in many countries, including the United States, Japan, China, and Spain. The target populations for the implementation of the systems in clinical practice include baby boomers in the United States, for HCV screening, and patients with hematological disorders, outpatients, and inpatients at hospitals, for HBV and HCV screening [14,15,16]. In the United States, alert systems improved the screening rate from 10% to 64% [15]. Konerman et al. [17] implemented an alert system for HCV screening in patients born between 1945 and 1965 in Ann Arbor; the screening rate increased from 7.6% prior to the alert system’s implementation to 72% 1 year after the alert system’s implementation. Alert systems not only increase the HCV antibody screening rate but also improve the rates of HCV RNA testing and referral. In Spain, Morales-Arraez et al. [18] used an automatic electronic alert message, increasing the HCV RNA testing rate from 62.4% to 77.7% and decreasing the time between RNA testing and positive HCV antibody tests from 19.1 ± 70.5 to 6.6 ± 10.0 months. We used an electronic reminder system to help clinicians identify patients with kidney disease whose HCV testing was unavailable at outpatient departments. The results show that this system can significantly improve the screening rate and not only the assessment rate of anti-HCV antibody but also the HCV RNA testing rate. After screening the 5538 patients without previous testing, 52 viremia patients with kidney disease were identified that can help to achieve the goal of HCV microelimination in this population.

The mean positivity rate for HCV antibodies in outpatients with kidney disease, excluding those with ESRD, was 5.7% in our hospital and was higher than that in the general population (4.1%) in the same county (Changhua) [19]. Although we know that uremic patients with ESRD in Taiwan (Kaohsiung) have a high anti-HCV seropositivity rate (13.6%) [20], no studies have focused on HCV screening in patients with kidney disease other than ESRD. The higher HCV rate in patients with kidney disease may be associated with chronic hepatitis C infection, which can induce insulin resistance and chronic inflammatory responses that have been linked to an increased risk of DM and CKD [3]. The higher HCV rate in patients with kidney disease still requires further study from different regions to confirm. The age distributions of HCV antibody-positive patients and HCV RNA-positive patients in our study were similar to those in other counties in Taiwan; higher HCV infection rates occurred in older patients, and no seropositive HCV antibody tests were observed in patients younger than 30 years in our study [21].

The anti-HCV seropositivity rate in the screening group was 2.5%, which was significantly lower than that in the baseline group (7.3%). The age distribution was similar between these two groups, and the baseline group was predominantly male (56.7%) compared with the screening group (51.4%) in our study. Some studies also revealed a lower anti-HCV antibody seropositivity rate after electronic medical record reminders [15]. Castrejón et al. [22] showed that the lower HCV antibody positivity in the screening population (1.5% vs. 4.1%, *p* < 0.01) may be associated with different sociodemographic makeup, selection criteria and racial/ethnic cohorts. Similar findings were also found in studies by MacLean et al. [23] (1.6% (intervention) vs. 7.0% (nonintervention), *p* < 0.001) and Sidlow and Msaouel [24] (0.86% (intervention) vs. 2.5% (nonintervention), *p* < 0.001). However, Federman et al. [25] showed inconsistent results: HCV antibody positivity was greater in the intervention group than in the control group (3.1% (intervention) vs. 1.1% (nonintervention), *p* < 0.0001). The lower HCV prevalence rate of the screening group in our study may be explained by the baseline group previously having more comorbidities or a higher rate of abnormal liver enzymes, leading to a high chance of receiving HCV testing before the study. The sex effect was not clear in our study because we found that the anti-HCV seropositivity rate was higher in females than in males (6.1% vs. 5.4%), but the HCV viremia rate was higher in males than in females (69.9% vs. 61%). The study of Wu et al. [26] showed that different gender predominances of HCV viremia may be associated with different genotypes and transmission patterns in Taiwan. The association between the effect of gender and HCV screening requires a further large-scale study in order to be confirmed.

The HCV viremia rate was also higher in the baseline group than in the screening group (69.1% vs. 46.8%). The FORMOSA-LIKE group by Yu-Ju Wei1 et al. [20] showed that the HCV viremia rate decreased after an 8-year follow up in uremic patients (73.8% to 56.3%) and that the HCV treatment rate increased from 2.3% to 21.7%. However, the treatment rate in patients without viremia was higher in the baseline group than in the screening group (11.6% vs. 5.1%) in our study, and the rate of anti-HCV signal-to-cutoff (S/CO) ratio ≥ 10.9 in patients without viremia was also higher in the baseline group than in the screening group (12.3% vs. 3.9%). A study by Seo et al. [27] revealed that age, serum alanine aminotransferase level, and anti-HCV S/CO ratio were associated with HCV viremia. The relationship between the ALT level or comorbidity and HCV viremia rate during HCV screening necessitates further study in the future.

There are some limitations to our study. First, we did not analyze the HCV prevalence and screening rate for each renal disease type. In our study, we included not only patients with a deterioration of renal function, such as those with chronic kidney disease stages 1–4, but also those with systemic diseases related to kidney diseases or structural abnormalities of the urinary system. Seropositivity rates of HCV in such populations are higher than those in the general population. To achieve HCV elimination by 2030, using an electronic reminder system to achieve microelimination in patients with kidney diseases is a feasible and efficacious method [28]. Second, our study did not analyze comorbidities, other biochemical data or fibrosis assessment in the baseline group, screening group or non-screening group. Further investigation is required to determine whether liver function tests and comorbidities would affect the screening results.

## 5. Conclusions

Using an electronic reminder system for HCV screening in patients with kidney disease is a feasible and useful method. This electronic reminder system can be considered as a hospital-based strategy to achieve HCV microelimination. A higher HCV prevalence rate in patients with kidney disease than in the general population and a lower HCV prevalence rate and viremia rate in the screening group were also found. The results require more large-scale studies to confirm this hypothesis in the future.

## Figures and Tables

**Figure 1 jcm-11-00423-f001:**
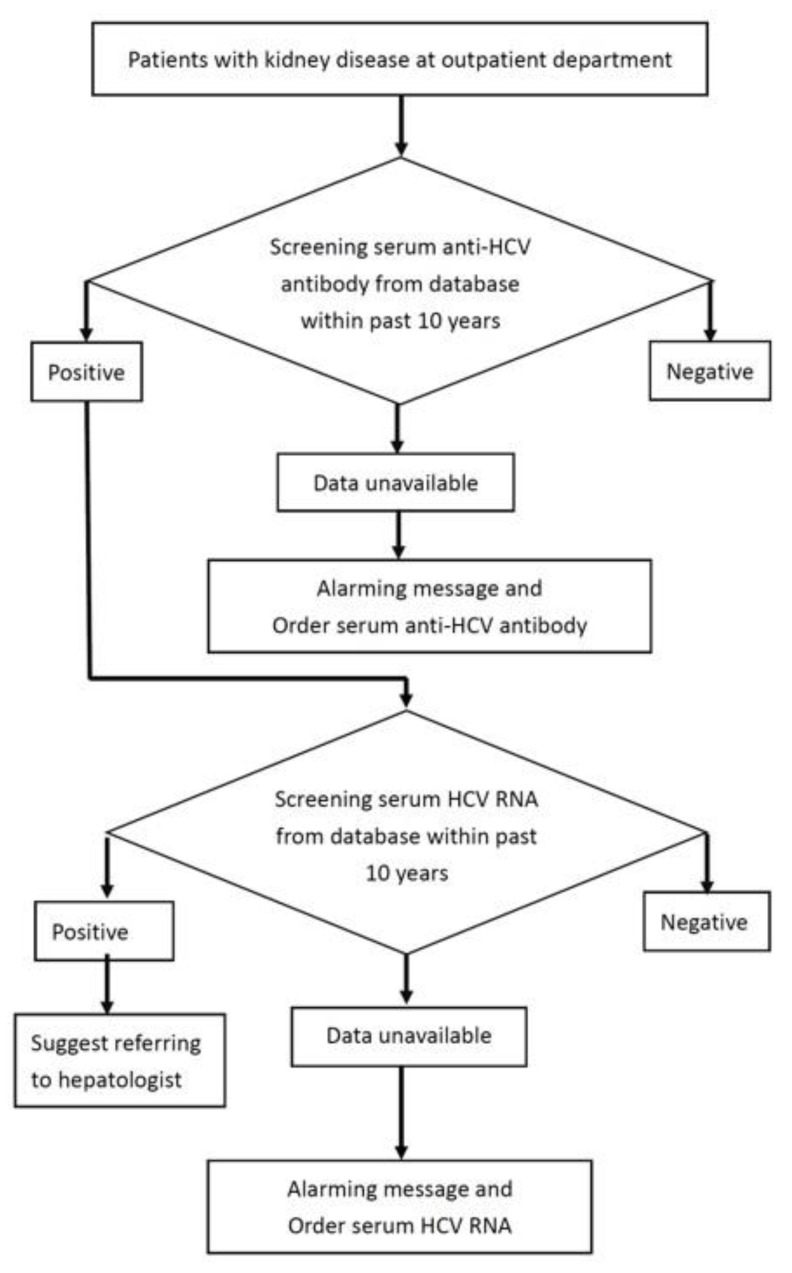
HCV screening flowchart targeting patients with kidney disease.

**Figure 2 jcm-11-00423-f002:**
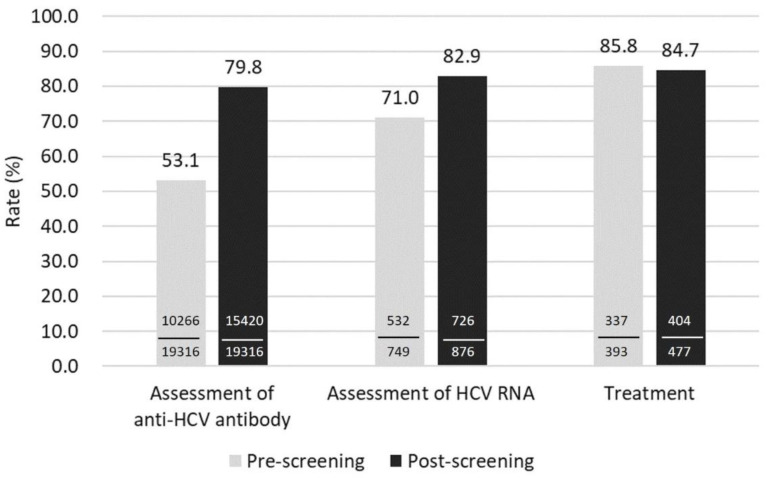
The rates of the assessment of anti-HCV antibody, HCV RNA, and treatment in the prescreening and post-screening periods.

**Figure 3 jcm-11-00423-f003:**
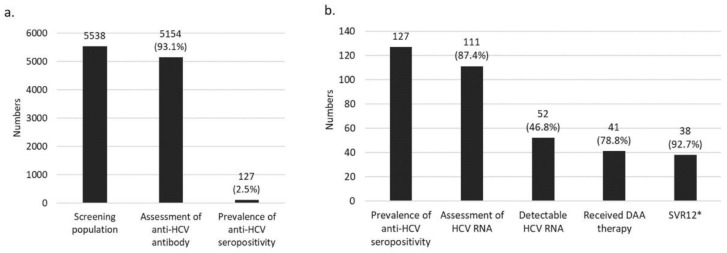
HCV care cascade. (**a**) Number and rate of anti-HCV antibody assessments. (**b**) Number and rate of HCV RNA detection, treatment and SVR12. * ITT-SVR was 92.7% and PP-SVR was 100%. The SVR data were unavailable in three patients until May 2021.

**Figure 4 jcm-11-00423-f004:**
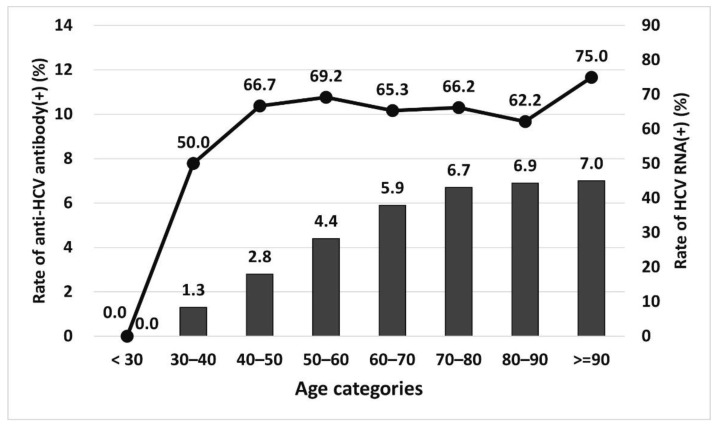
The prevalence of seropositive HCV antibody and HCV viremia in each age category.

**Table 1 jcm-11-00423-t001:** The basic characteristics, anti-HCV seropositivity rate and percentage of patients with detectable HCV RNA among the baseline, screening, and total population groups. The baseline group was defined as patients who had laboratory tests before screening, and the screening group was defined as patients who completed tests during the screening period.

	Total	Baseline	Screening	*p* Value ^a^
Age	67.1 ± 14.1	67.1 ± 14.3	67.0 ± 13.7	0.744
Sex (male)	54.90%	56.70%	51.40%	<0.001
Anti-HCV seropositive	876/15,420 (5.7%)	749/10,266 (7.3%)	127/5154 (2.5%)	<0.001
Detectable HCV RNA	477/726 (65.7%)	426/615 (69.1%)	52/111 (46.8%)	<0.001

Data were analyzed using chi-square tests. ^a^ *p* value for baseline vs. screening groups.

## Data Availability

Not applicable.

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
