# Peer review of "Micro-Elimination of Hepatitis C among Patients with Kidney Disease by Using Electronic Reminder System—A Hospital-Based Experience"

_jcm, 2022, doi:10.3390/jcm11020423_

Round 1

Reviewer 1 Report

The manuscript presents a study evaluating the usefulness of the electronic reminders system in HCV infection screening in patients with kidney diseases. The study is interesting, carried out on a large group of patients, and provides valuable data to improve the detectability of the infection in this specific group of patients.

The study also has some limitations:

Understanding the essence of the study from the abstract is somewhat difficult and this part of the manuscript would require revision (e.g. “We tried to determine whether reminder systems could improve the HCV screening rate and the HCV prevalence rate before and after screening in such populations”. What populations did the authors mean?)

The site where the study was conducted isn’t sufficiently clearly stated in the text. The authors use the phrase "at our hospital" instead of specifying its name. It is unclear because the affiliation list includes 2 different research sites.

The quality of the manuscript language would gain value if it was improved in cooperation with a native speaker or an editorial service

Author Response

Comments from reviewer 1

Point 1: Understanding the essence of the study from the abstract is somewhat difficult and this part of the manuscript would require revision (e.g. “We tried to determine whether reminder systems could improve the HCV screening rate and the HCV prevalence rate before and after screening in such populations”. What populations did the authors mean?)

Response 1: Thank you for your kind suggestion. The abstract has been rewritten to help the readers can understand the essence more easily. (Line 15-22, page 1)

Point 2: The site where the study was conducted isn’t sufficiently clearly stated in the text. The authors use the phrase "at our hospital" instead of specifying its name. It is unclear because the affiliation list includes 2 different research sites.

Response 2: Thank you for your kind suggestion. The phrase "at our hospital" has been changed to “Changhua Christian Hospital”. (Line 62, 72, page 2)

Point 3: The quality of the manuscript language would gain value if it was improved in cooperation with a native speaker or an editorial service

Response 3: Thank you for your kind suggestion. The manuscript has been re-edited by MDPI English Editing.

Reviewer 2 Report

This is an interesting manuscript from Pei-yuan Su that comes to complement their recently published work in implementing an electronic reminder system in at-risk populations having previous medical conditions that are associated with higher HCV prevalence.

The study is interesting and in line with current recommendations to increase micro-elimination efforts, in the bid to end HCV by 2030. It shows how this strategy may work in a medical system which has a fully functional electronic records system, as well as an efficient primary care medical body.

Regarding the methodology, one thing that is not clear would be if patients were included irrespective of the duration of their pre-existing kidney condition and hemodialysis, since the procedure itself may be a risk factor for contracting HCV. 

Minor spelling and clarity improvements are also needed.

Author Response

Comments from reviewer 2

Point 1: Regarding the methodology, one thing that is not clear would be if patients were included irrespective of the duration of their pre-existing kidney condition and hemodialysis, since the procedure itself may be a risk factor for contracting HCV.

Response 1: Thank you for your kind suggestion. The study included patients according to diagnoses by ICD-10-CM codes from previous medical records in the period of January 1st, 2018, to December 31st, 2018. The duration of their pre-existing kidney condition is not recorded, and hemodialysis has been excluded in the study. (Line 60-61, page 2)

Point 2: Minor spelling and clarity improvements are also needed.

Response 2: Thank you for your kind suggestion. The manuscript has been re-edited by MDPI English Editing. (English Editing ID: english-38792)